# The Clinical Role of miRNAs in the Development and Treatment of Glioblastoma

**DOI:** 10.3390/ijms26062723

**Published:** 2025-03-18

**Authors:** Samantha Epistolio, Paolo Spina, Ismail Zaed, Andrea Cardia, Francesco Marchi, Milo Frattini

**Affiliations:** 1Institute of Pathology, Ente Ospedaliero Cantonale (EOC), 6900 Locarno, Switzerland; samantha.epistolio@eoc.ch (S.E.); paolo.spina@eoc.ch (P.S.); 2Service of Neurosurgery, Neurocenter of the Southern Switzerland, Regional Hospital of Lugano, EnteOspedaliero Cantonale (EOC), 6900 Lugano, Switzerland; ismaiel.zaed@eoc.ch (I.Z.); andrea.cardia@eoc.ch (A.C.); francesco.marchi@eoc.ch (F.M.)

**Keywords:** GBM, miRNAs, prognosis, survival, treatment, TMZ

## Abstract

Glioblastoma multiforme (GBM) is the most common brain tumor and one of the most aggressive, with a median overall survival (OS) of only 15–18 months. These characteristics make it necessary to identify new targets for the improvement of prognosis and better prediction of response to therapies currently available for GBM patients. One possible candidate target could be the evaluation of miRNAs. miRNAs are small non-coding RNAs that play important roles in post-transcriptional gene regulation. Due to their functions, miRNAs also control biological processes underlying the development of GBM and may be considered possible targets with a clinical role. This narrative review introduces the concept of miRNAs in GBM from a clinical and a molecular perspective and then addresses the specific miRNAs that are most described in the literature as relevant for the development, the prognosis, and the response to therapies for patients affected by GBM.

## 1. Introduction

Each year, approximately 85,000 people in the United States are diagnosed with a primary brain tumor, with about 29% of these being malignant [1]. In adults, 80% to 85% of these are gliomas. The fifth edition of the WHO Classification of Central Nervous System (CNS) Tumors identifies these entities as “adult-type diffuse gliomas” [2]. Among this group, glioblastoma (GBM) stands out as the most common malignant primary brain tumor in adults, and it is the leading cause of death in patients with primary brain tumors. GBM typically appears after the age of 40, peaking in incidence among those aged 75 to 84 years [3]. Survival rates for GBM decrease with age: only 5% of all diagnosed individuals survive for five years, and this figure falls to 2% for patients aged 65 years and older [4]. Despite advancements in understanding the biology of these tumors, significant improvements in treatment and patient outcomes are still needed.

## 2. General Molecular Features

Fewer than 5% of adults diagnosed with malignant brain tumors report a family history of brain tumors or a predisposition syndrome for malignancies. Around 5% of all gliomas are considered to be familial, with several rare Mendelian inherited syndromes associated with GBM and other gliomas [5]. Interestingly, the presence of germline variants appears to be higher than anticipated, with up to 13% of glioma patients carrying at least one harmful or potentially harmful genetic mutation in their germline [6].

The causes of most GBMs remain largely unidentified. While gliomas have been experimentally induced in rats using chemical carcinogens like ethylnitrosourea and methylnitrosourea, there is no strong evidence that these chemicals contribute to glioma development in humans. Similarly, although genome sequences and proteins from polyomaviruses (e.g., SV40, BK virus, and JC virus) have been detected in gliomas, their presence was infrequent in a more recent large series. A small percentage of GBMs occur in multiple members of the same family or are linked to inherited genetic tumor syndromes, such as Lynch syndrome, constitutional mismatch repair deficiency syndrome, Li–Fraumeni syndrome, and neurofibromatosis type 1. Genome-wide association studies have uncovered genetic variants in genes like *TERT*, *EGFR*, *CCDC26*, *CDKN2B*, *PHLDB1*, *TP53*, and *RTEL1* that are associated with an elevated risk of gliomas. Additionally, specific single-nucleotide polymorphisms (SNPs) have been identified as increasing glioma risk, distinct from those linked to other types of brain tumors.

The rising incidence of GBM suggests a potential role for environmental factors, but despite extensive research, most environmental exposures, including non-ionizing radiation (e.g., from mobile phones) and occupational hazards, have not been conclusively linked to the disease. The only confirmed environmental risk factor is exposure to ionizing radiation targeting the head and neck. For instance, patients treated for acute lymphoblastic leukemia, or atomic-bomb survivors, are at greater risk of GBM, though no association has been found with diagnostic radiation. Interestingly, individuals with a history of allergies or atopic conditions appear to have a reduced risk of developing the disease [2].

## 3. Glioblastoma Management

The current management of GBM, one of the most aggressive and lethal brain tumors, necessarily involves a comprehensive, multidisciplinary approach. Despite the fact that 20 years have passed since its first publications, the Stupp Protocol, which foresees a combined chemo-radiation therapy, remains the gold-standard treatment [7]. Initially, the treatment strategy centers on maximally safe surgical resection, aiming to remove as much of the tumor as possible while preserving neurological function. This surgery helps to reduce the tumor burden and alleviate symptoms.

Following surgery, patients typically undergo radiotherapy combined with chemotherapy using temozolomide (TMZ). Radiotherapy targets residual cancer cells, while TMZ, an oral alkylating agent, enhances the cytotoxic effects of radiation and has shown to improve overall survival (OS) rates [8,9].

Despite aggressive treatment, GBMs are notorious for their high recurrence rates and, consequently, for the severe and short course. As a consequence, research is increasingly focused on novel therapies and combination treatments. Tumor-treating fields (TTFs), a non-invasive treatment that uses alternating electric fields to disrupt cancer cell division, has emerged as a promising adjunct to standard therapy, extending survival in some patients [10].

Moreover, advancements in molecular and genetic profiling of GBMs are paving the way for personalized medicine [11]. Identifying specific genetic mutations and molecular markers allows for tailored treatment strategies, potentially improving efficacy and reducing side effects. Immunotherapies, including immune checkpoint inhibitors and CAR-T cell therapy, are under investigation, aiming to harness the body’s immune system to recognize and destroy cancer cells [12].

Clinical trials continue to explore the potential of targeted therapies, such as inhibitors of specific signaling pathways involved in GBMs progression. Bevacizumab, an anti-angiogenic agent, has been used to manage symptoms and prolong PFS, although its impact on OS remains unclear [13].

Despite these advancements, GBM management remains challenging, with a median survival time of approximately 15–18 months post-diagnosis. The aggressive nature of the tumor, its location within the brain, and its genetic heterogeneity contribute to its poor prognosis. Continuous research and clinical trials are crucial in the ongoing effort to discover more effective treatments and improve patient outcomes.

## 4. Clinical Relevance of the Molecular Profile

The rise in both molecular profiling and the use of machine learning techniques has led to more accurate prognostic assessments and personalized treatment strategies. Discovering new alterations within GBM presents opportunities for developing targeted drugs, while correlating these genic dysregulations with specific clinical courses improves our ability to diagnose and predict disease severity.

In 2017, the establishment of the Consortium to Inform Molecular and Practical Approaches to CNS Tumor Taxonomy (cIMPACT-NOW) was announced. This consortium aims to evaluate and suggest updates to the WHO classification of brain tumors, incorporating the latest molecular insights and their practical implications for clinical practice [14].

The updated WHO criteria and nomenclature introduced in 2021 have placed a stronger emphasis on the role of molecular genetics in diagnosing GBM:

Astrocytoma, IDH-mutant: Previously, IDH-mutant tumors were classified as diffuse astrocytoma, anaplastic astrocytoma, or GBM. The latest classification now consolidates these into a single type of IDH-mutant astrocytoma, graded as 2, 3, or 4.Grading criteria: The grading of IDH-mutant diffuse astrocytic tumors is no longer based solely on histology. It also considers the presence of the *CDKN2A/B* homozygous deletion mutation, which results in a CNS WHO grade of 4, even if microvascular proliferation or necrosis is absent.GBM, IDH-wildtype: This classification identifies specific molecular markers for this tumor, such as the presence of a *TERT* promoter mutation (associated with increased telomerase activity, crucial for tumor cell immortalization) or *EGFR* gene amplification, leading to overexpression of the receptor and the combined gain of chromosome 7 and loss of chromosome 10 (+7/−10). If any of these markers are found in an IDH-wildtype diffuse astrocytic glioma in adults, the diagnosis should be GBM, IDH-wildtype.Pediatric patients: The diagnostic criteria for IDH-wildtype diffuse astrocytomas differ in pediatric patients, who are diagnosed using different categories of pediatric-type gliomas [2].

IDH-wildtype GBMs are defined by the absence of mutations in *IDH1* codon 132 and *IDH2* codon 172, as well as the absence of mutations in H3 p.K28X (formerly p.K27X) or H3 p.G35X (formerly p.G34X) [15]. For patients aged 55 years or older at diagnosis, with a histologically classic GBM that is not located in midline structures and has no history of pre-existing lower-grade glioma, the lack of immunoreactivity for IDH1 p.R132H is sufficient to diagnose IDH-wildtype GBM [16]. In such cases, further DNA sequencing is unnecessary, as the probability of non-canonical *IDH* mutations is very low (less than 1%) in GBMs of patients aged 55 years or older [17].

However, for patients younger than 55 years, or those with a history of lower-grade glioma and/or tumors showing immunohistochemical loss of nuclear ATRX expression, negative IDH1 p.R132H immunostaining should be followed by DNA sequencing to identify less common *IDH1* or *IDH2* mutations. If no *IDH* mutations are found through sequencing, these tumors are classified as GBM, IDH-wildtype [18].

Tumors located in midline structures should be evaluated for the H3 p.K28M mutation (formerly known as p.K27M) to exclude diffuse midline glioma, H3 p.K28X–altered. In hemispheric tumors, especially in younger patients, it is important to rule out H3 p.G34X–mutant diffuse hemispheric gliomas by using immunohistochemistry for H3.3 p.G35R (formerly p.G34R) or H3.3 p.G35V (formerly p.G34V) mutations or through sequencing of H3-3A (H3F3A) [15]. This comprehensive approach ensures accurate diagnosis and proper classification of GBMs based on the presence or absence of specific genetic mutations.

The presence of at least one of these molecular aberrations in an IDH- and H3-wildtype diffuse glioma is sufficient for the diagnosis of IDH-wildtype GBM, even in the absence of microvascular proliferation and/or necrosis [19].

In addition to these genetic alterations, DNA methylation profiling has emerged as a valuable tool for diagnosing and stratifying GBMs. A significant calibrated score for the DNA methylation profile of IDH-wildtype GBM is sufficient for diagnosis. DNA methylation profiles can further classify GBMs into molecular subgroups, with the RTK1, RTK2/classic, and mesenchymal subgroups being the most common in adult patients [20]. While the clinical relevance of methylation-based subgroups in adult patients is somewhat limited, high-grade gliomas in children and adolescents may exhibit less common DNA methylation profiles that are associated with significantly longer survival [21].

Overall, DNA methylation profiling can assist in diagnosing challenging cases and differentiating GBM from histologically similar entities, enhancing our understanding of the molecular diversity within this aggressive brain tumor [20].

Molecular profiling has enabled researchers to identify common genetic mutations and core pathways shared among sporadic GBM, leading to the identification of three primary subgroups:

Proneural group: This group is characterized by proneural gene expression patterns and RTK I/LGm6 DNA methylation profiles. This subgroup often shows amplifications of genes such as cyclin-dependent kinase 4 (*CDK4*) and platelet-derived growth factor alpha (*PDGFRA*). It is more prevalent among younger adults.Classical group: This group exhibits classical gene expression patterns and classic-like RTK II DNA methylation profiles. It is marked by frequent *EGFR* amplifications and the loss of *CDKN2A/B* genes.Mesenchymal group: This group OS enriched for tumors with neurofibromatosis type 1 (*NF1*) loss and increased infiltration by macrophages [22]. This subgroup is associated with a mesenchymal or mesenchymal-like subtype.

These subgroups are defined by specific genetic alterations and provide insights into the molecular diversity of GBMs.

These three distinct subgroups, along with mixed entities that exhibit characteristics from multiple subgroups, encompass the majority of GBMs. While molecular classification has provided valuable insights and a framework for research, its clinical utility in GBM treatment remains uncertain. None of the GBM subtypes can reliably predict responses to current therapies. Moreover, assigning a specific subtype can be challenging, as tumors may display mixed subtypes simultaneously, and subtype characteristics may evolve throughout the disease course [20].

Currently, the sole predictive biomarker for treatment response to TMZ is the status of *MGMT*-mediated DNA repair silencing. This silencing typically results from *MGMT* promoter methylation and loss of the second allele of chromosome 10 [23,24].

*MGMT* promoter methylation status is routinely assessed in IDH-wildtype GBMs because it provides crucial clinical information regarding chemotherapy response and patient survival [25]. Specifically, it helps predict how patients will respond to treatment with drugs like TMZ or TMZ plus lomustine (CCNU) [26,27]. In elderly patients, *MGMT* promoter methylation status can guide treatment decisions, helping determine whether chemotherapy or radiotherapy is more appropriate [28,29].

Furthermore, the presence of *IDH1/2* mutations in adult diffuse gliomas is associated with prolonged patient survival. A rapid and cost-effective initial screening method for *IDH* mutation is mutation-specific immunohistochemistry, particularly for the common variant IDH1 p.R132H, which accounts for the vast majority of *IDH* mutations in GBM. A positive result in this test confirms the presence of an *IDH* mutation [2].

For cases where initial immunohistochemistry is negative (i.e., “antibody-negative” GBM), further testing, such as targeted sequencing, may be considered. However, the decision for additional testing depends on various factors, including the patient’s age. Non-canonical *IDH* mutations are extremely rare in older patients (over 55 years). Additionally, GBMs with microthrombi and/or clear pseudopalisading necrosis at initial diagnosis are highly unlikely to harbor an *IDH* mutation [30].

Genomic profiling has significantly advanced our understanding of the molecular mechanisms underlying GBM and identified potential avenues for developing targeted therapies tailored to specific patient groups or for identifying specific morphologic patterns. Particularly, *BRAF* p.V600E mutation, relatively rare in IDH-wildtype GBMs, is detectable in up to 50% of GBMs with epithelioid histology. It is highly prevalent (79%) in pleomorphic xanthoastrocytoma-like tumors and 35% of adult-type IDH-wildtype GBMs, but it is absent in pediatric RTK1 tumors [31].

Despite providing valuable insights into tumor biology and heterogeneity; however, gene expression profiles have not yet gained clear significance in routine clinical diagnostics for GBM [14].

It is important to note that none of the genetic alterations mentioned are specific or sufficient for defining the respective morphological subtypes or patterns of GBMs. The molecular characterization of these tumors is complex, and individual GBMs can exhibit multiple genetic changes, especially after chemo-radiation therapies. Following initial treatment, indeed, which typically involves surgical resection, radiation therapy, and chemotherapy, distinct subgroups of tumor cells may emerge with unique characteristics. For example, approximately 10% of recurrent GBMs, occurring after treatment with TMZ, exhibit significantly higher mutation rates [32]. This phenomenon of DNA “hypermutation” is often associated with underlying genetic deficiencies in DNA mismatch repair (MMR) genes. Additionally, hypermutation can develop due to exposure to DNA alkylating agents, particularly in gliomas with *MGMT* methylation, including those with *IDH* mutations [33].

Moreover, comparisons between tumor samples obtained at diagnosis and those from recurrence show that approximately 80% of mutations and copy-number variations remain consistent between primary and recurrent tumors. Genetic events like mutations in *PIK3CA*, *TERT* alterations, and *EGFR* amplifications found in the primary tumor typically persist in the recurrent tumor. Conversely, events such as *PDGFRA* amplifications, *EGFR* mutations, and the presence of *EGFR* variant III (*EGFRvIII*) rearrangement are more likely to be lost. Commonly acquired genetic changes in recurrent tumors include mutations in *TP53*, *EGFR*, and phosphatase and tensin homolog (*PTEN*) [34].

Emerging sequencing technologies provide deeper insights into intratumoral heterogeneity and the evolution of GBMs. Single-cell transcriptomics have revealed that GBMs contain cells representing each of the three gene expression subtypes, rather than fitting neatly into a single category. This supports earlier findings from bulk gene expression profiling across multiple tumor sectors.

Moreover, sequencing circulating tumor DNA (ctDNA) present in cerebrospinal fluid (CSF) can provide a genetically accurate snapshot of the glioma genome in up to 50% of patients. This advancement may potentially obviate the need for repeat tumor biopsies in certain cases. Ongoing technological advancements may also make it feasible to evaluate plasma ctDNA in the future.

Further research is exploring novel predictive biomarkers for molecularly targeted therapies in subsets of GBM patients. Promising biomarkers include high tumor mutation burden, *BRAF* p.V600E mutation, *NTRK* or *FGFR* gene family fusions, and *MET* amplification or fusions. These markers hold potential for tailoring therapeutic approaches in specific GBM subgroups and improving treatment outcomes [35].

## 5. microRNAs (miRNAs) and GBM

### 5.1. miRNAs Function and Biogenesis

miRNAs are small non-coding molecules of RNAs of the length of 19–23 nucleotides that exert post-transcriptional regulatory effects on many genes. Their action is based on a process of silencing of the mRNAs defined as RNA interference (RNAi). RNAi can cause mRNA degradation or sequence-specific mRNA inhibition [35]. It has been estimated that nearly 30% of genes in the human genome are regulated by miRNAs [36,37].

miRNAs derive from primary transcripts (pri-miRNAs), which are transcribed from the genome by the RNA polymerase II. In the nucleus, the pri-miRNAs are then modified, with the 5′ cap and the 3′ polyadenylation tail, in 60–70 nucleotides pre-miRNA by the RNAse III-Drosha complex. Subsequently, pre-miRNAs are exported in the cytoplasm through the nuclear transporter exportin 5. To conclude the process of miRNA biogenesis, in the cytoplasm, pre-miRNAs are cut and separated into two strands of 19–23 nucleotides by the Dicer protein, an endoribonuclease. These strands form a duplex where the antisense strand is incorporated into the RISC complex (developing the miRNA-induced silencing complex, miRISC), while the sense strand is degraded. The complementary attachment of the miRISC complex to the specific mRNA in the 3′ untranslated region of mRNA can bring about mRNA degradation if there is a high degree of complementarity, or it can lead to the inhibition of mRNA if the complementarity is partial [38,39] (Figure 1).

### 5.2. miRNAs in Cancer

Since miRNAs are involved in the regulation of a variety of genes, they are also implicated in several pathological conditions, especially cancer. In the development of tumors, they can act like oncogenes (oncomiRs) or as tumor-suppressor genes [40]. OncomiRs are upregulated and promote cancer through inhibition of the genes involved in the control of cell differentiation, apoptosis, and/or through the negative regulation of tumor suppressor genes. Instead, the miRNAs acting like tumor-suppressor genes are downregulated, and they can act through the regulation of the genes for cell differentiation or apoptosis, or through regulation of oncogenes [40]. The evaluation of the expression of miRNAs’ signature can be a diagnostic tool for the definition of cancer disease stage and survival. In addition, in some cases, it can help for the decision of a tailored treatment based on the specific characterization of miRNAs’ expression status [41]. In particular, miRNA expression disorders in cancer affect cell proliferation, invasiveness, angiogenesis, and recurrence [42,43]. Their role in tumorigenesis is further demonstrated by the fact that their genes are located in general near chromosomes’ fragile sites, which are known to be more susceptible to amplifications, to deletions, to point mutations, and even to DNA methylation disorders [44,45].

### 5.3. miRNAs in GBM

Recent years have seen an increase in publications focusing on the function of miRNAs in GBM, demonstrating how these molecules can influence tumor development, prognosis, response to therapies, and, in some cases, even the classification of this cancer [35,46,47]. Indeed, miRNAs in GBM can influence the promotion of sustained signaling of proliferation or evasion from tissues [47]. In a tumor, the expression levels of miRNAs dynamically change at the various stages [47].

In this section, we report a list and the description of some of the most relevant miRNAs involved in GBM; a summary is reported in Table 1 and Table 2. In this list, the miRNAs are subdivided between the ones that act like oncomiR (Table 1) (Figure 2) or like tumor-suppressor miRNAs (Table 2) (Figure 3) in GBM development. We will describe the miRNAs for which at least three publications associating them with GBM have been published to date in order to cover the largest, but also the most reliable, area of this topic, without dwelling on the miRNAs with a really minimal number of publications (one or studies).

#### 5.3.1. OncomiR-Upregulated miRNAs

##### OncomiR Involved in GBM Biogenesis

miR-17-92 cluster

This miRNA cluster regulates, through inhibition, the tumor-suppressor genes that act as cell-cycle inhibitors (e.g., *PTEN* and *CDKN1A*), thus promoting GBM Stem Cell (GSC) differentiation and blocking apoptosis (Figure 2).

The overexpression of this miRNA cluster is higher in GBM tumor samples and cell lines than in normal brain tissues. High levels of miR-17-92 cluster are associated with high aggressiveness of the tumor because the expression of this cluster is associated with higher invasion and replication capability [48,49]. Recently, focusing only on miR-17 alone in GSCs, the opposite has been demonstrated: miR-17 is downregulated in differentiated GSCs cell lines. This also demonstrates that miR-17 alone has a potential regulator role in the process of GSC/GBM cell transition, and this role is different from that of the cluster miR-17-92 [50].

##### OncomiR Involved in GBM Prognosis

miR-9

miR-9 promotes GBM tumor cell proliferation and inflammation, targeting markers of the extracellular matrix and proteins involved in cell duplication and transformation, including RAS and MYC [45,46] (Figure 2). miR-9 activates oncogenic *RAS*, which induces VEGF expression, and VEGF expression is able to potentiate MYC activity to promote tumor angiogenesis [51,52].

In addition, when upregulated, miR-9 downregulates the *PTCH1* tumor-suppressor gene, resulting in the activation of Sonic HedgeHog (SHH), a protein with neuronal morphogenic activity that, when activated, increases the cell proliferation signal [51] (Figure 2). In GBM cell lines that model the expression of this oncomiR brought to resistance to the standard therapy based on TMZ, this could be associated with the PTCH1-SHH pathway. Indeed, in TMZ-resistant cells, PTCH1 mRNA was significantly (*p* < 0.05) increased, but the protein was decreased, suggesting that TMZ could induce post-transcriptional regulation of PTCH1 [53]. Due to TMZ resistance in vitro, miR-9 expression has been evaluated in patients, comparing short-term and long-term survivors, and it was found that it is associated with poor survival [51,53,54]. To conclude, a study by Geng et al. described a possible application of miR-9 as a future novel diagnostic biomarker, demonstrating that the expression level of miR-9 was significantly higher in GBM extracellular vesicles derived from cerebrospinal fluid and tissues than normal tissue [55].

miR-10a and miR-10b

miR-10a and miR-10b block cell-cycle inhibitors’ activity (such as CDKN1A, BIM, BCL2, TEAP2C, and PTEN) and HOXD10, a protein involved in differentiation [56,57] (Figure 2). As a consequence, these oncomiRNAs promote cell proliferation, invasion, migration, and epithelial–mesenchymal transition (EMT), in addition to having an effect on GSCs [56,57,58,59]. The expression of miR-10b has been associated with the degree of the brain tumor being higher in GBM than in other gliomas [60,61,62].

For what concerns the treatments based on this marker, hypothetically, the development of a drug able to block upregulation of miR10 should restore the correct target genes’ expression and could reduce GBM cell growth [47]. Recently, the relevance of the synergistic action between miR-10b and miR-222 has also been described. These two miRNAs promote cell proliferation, blocking PTEN, which activates p53. Additionally, this pair of miRNAs regulates apoptosis, modulating BIM, an apoptotic factor. As a matter of fact, they have been associated with poor survival, and due to their joint effect on GBM development, in the future, both of these miRNAs will be considered as putative new targets for the treatment of GBM [63].

miR-148a

The main target of miR-148a is cell adhesion molecule 1 (CADM1). The inhibition of CADM1 leads to an increase in the STAT3 pathway [56,64] (Figure 2). Another target of miR-148a is the factor inhibiting HIF-1 (FIH1) that through inhibition of HIF-1, a regulator of homeostasis, is able to influence angiogenesis [65,66].

This miRNA has been included by the The Cancer Genome Atlas (TCGA) on the list of miRNAs involved in GBM development. The first fact justifying this classification is that miR-148a has been found to be overexpressed in plasma from the serum of patients with GBM if compared to healthy cases [64]. The second one is the role in GBM. In a 2011 study, Srinivasan and colleagues identified a signature of ten miRNAs whose expression is associated with GBM patient survival [65]. One of these miRNAs is miR-148a, which has been found to be expressed in the high-risk group (i.e., patients characterized by low survival). The relevant role of this miRNA in high-risk GBMs is justified by its aforementioned biological activity of angiogenesis regulation, leading to stimulation of the invasion of the tumor [56,64,65].

miR-182

The main targets of miR-182 are BCL2L12, HIF2A, MET, CYLD, and LRRC4, leading to uncontrolled proliferation, stemness, and, in contrast, also to apoptosis [47,67] (Figure 2).

The expression of this miRNA is associated with the molecular status of brain tumor malignancy. In particular, the cluster miR-183/96/182, including miR-182, has been reported to be upregulated in tumors with *EGFR* amplification [68,69,70]. To date, the influence of this miRNA on GBM survival is still debated. In many publications the expression of miR-182 correlates with a better response to TMZ-based chemotherapy and, as a consequence, with better survival. In particular, Zhao et al. found that three serum miRNAs (miR-182, 15b-5p, and 145-5p), including miR-182, are significantly associated (*p* = 0.04) with 2-years OS, and Kouri et al. reported how this miRNA is detected at high levels in patients with better survival (*p* = 0.01) [71,72]. This could be due to the pro-apoptotic effect and to the downregulation of cancer cell migration [71,72]. On the other side, some research groups have described the effect of miR-182 on growth promotion and on migration and, consequently, have associated its expression with worse prognosis [73,74]. In particular, it has been found that the cumulative 5-year OS rate is lower when miR-182 expression is high than in the low-miR-182 group for what concerns OS (*p* = 0.003) and disease-free survival (DFS) (*p* = 0.006) [73]. It could be hypothesized that this difference between studies is due—more than to methodology, which is comparable—to an ethnic difference between study populations.

miR-196a and miR-196b

These miRNAs target different markers associated with proliferation and with the regulation of apoptosis, such as HOXB8, HOXC8, HOXD8, HOXA7, HOXB7, ERG, HMGA2, and ANXA1 [75,76]; as a consequence, when miR-196 is overexpressed in glioma cells, proliferation is induced [76,77] (Figure 2). A study published by Lakomy et al. in 2011 demonstrated how this miRNA can influence sensitivity to TMZ [78]. Indeed, miR-196b was correlated in a positive way with OS (*p* = 0.0492) [78]. In addition to the results of this group, the only ones published so far, our group, in 2023, obtained the same results but in a larger and consecutively enrolled cohort [79]. In particular, we found a significant correlation between GBM OS and miR-196b expression (*p* = 0.008), where a better OS was associated with miR-196 expression. In our study, the same data were not statistically significant concerning progression-free survival (PFS) [79].

##### OncomiR with Effect on Therapy Efficacy

miR-26a

miR-26a binds to PTEN and ATM, causing their inhibition and the promotion of tumor formation. In vitro (testing U87 cells, a cell line from malignant glioma), it has been reported that the overexpression of this miRNA, reducing DNA repair ability, enhances radiosensitivity to radiotherapy [57,80,81,82] (Figure 2). In addition, the knock down of this miRNA acts in the opposite way, reducing this sensitivity [81]. From the molecular point of view, the response to radiotherapy is modulated by this miRNA through the targeting of the 3′UTR region of the ATM gene, leading to reduced ATM levels and to the consequent inhibition of the homologous recombination repair system. In the future, these in vitro studies could be expanded on patient tissues to clarify whether miR-26 could play a role as a radiosensitizer in GBM. However, to date, in vivo results on GBM patients are not yet available.

miR-648

miR-648 is involved in the regulation of Myelin-associated oligodendrocyte basic protein isoform 1 (MOBP) [83] (Figure 2) and has been associated with the low expression of miR-181c, miR-181d, and miR-195 [84]. miR-648 is a member of a group of miRNAs that have been found, by a bioinformatically guided experimental approach by Kreth et al., to be capable of downregulating MGMT expression independently of promoter methylation via elongation of the 3′-UTR end of the MGMT mRNA [85]. In particular, this research group described how this miRNA is able to regulate MGMT at the post-transcriptional level [85], and, only in cell lines, they observed that the expression by transfection of miR-648 enhanced the responsivity of TMZ in MGMT-expressing T98G glioma cells [85]. To the best of our knowledge, until now, few data have been obtained concerning the association between miR-648 and TMZ response in samples from patients. A study published by our group in 2023 reported a correlation between OS and miR-648 expression (*p* = 0.02): the high expression of this miRNA correlates to better survival but not PFS in GBM treated with TMZ according with the Stupp scheme (60 Gray radiotherapy and concomitant chemotherapy with TMZ, followed by six cycles of maintenance with TMZ) [79]. To conclude, in 2024, our research group found that the low expression of miR-648 and miR-181c or miR-181d pairs is associated with worse prognosis than cases with other low-expressed miRNA pairs (miR-21, miR-195, and miR-196b) [84].

##### OncomiRs Involved in GBM Biogenesis That, in the Future, Will Have a Clinical Role Through Their Inhibition

miR-21

miR-21 is overexpressed in many tumors, including GBM, in which its role is to regulate the inhibition of PTEN and p53, and the activation of EGFR, Cyclin D1, and AKT2 (Figure 2) [86]. Other targets of miR-21 are SPOCK1, a proteoglycan, and transcription regulators such as RB1CC1 [35]. In addition, miR-21 enhances not only pro-proliferative action but also tumor invasion and migration, targeting the factors that regulate matrix metalloproteinase (e.g., RECK and TIMP3). Historically, this miRNA was the first to be described as an oncogenic miRNA able to contribute to the progression of GBM. Based on the overexpression of these miRNAs, it has been suggested that, in the future, different silencing mechanisms could be implemented in order to improve the chance of treatment of GBM [48,56,59]. These different mechanisms are generally based on RNA interference and could be of two types: genome-derived miRNAs and exogenous miRNAs transferred inside exosomes. However, the mechanisms of inhibition are still at the drawing stage, and data have not been published so far.

#### 5.3.2. Tumor-Suppressor miRNAs-Downregulated miRNAs

##### Tumor-Suppressor miRNAs Involved in GBM Biogenesis

miR-1

This miRNA binds connexin-43 and can target glucose-6-phosphate dehydrogenase (G6PD), causing the inhibition of tumor cell proliferation and of tumor cell migration [56,87,88] (Figure 3). Some in vitro experiments demonstrated that miR-1 inhibition can enhance the sensitivity of GBM cells toward TMZ, and, when downregulated, it increases tumorigenesis, leading to cell proliferation [56,87,88]. In particular, it has been described how the expression of miR-1 in GBM cell lines targets fibronectin. Subsequently, it has been observed that, when fibronectin is highly expressed in GBM, there is poor patient survival [56]. Liu et al. demonstrated that in GBM cell lines, miR-1-3p is significantly downregulated compared to normal cell lines (*p* < 0.01) [89]. To date, to the best of our knowledge, no additional data have been obtained for tumor tissues.

##### Tumor-Suppressor miRNAs Involved in GBM Prognosis

miR-128

This miRNA has several targets in the apoptosis and cell growth pathways: WNT, ERK, EGFR, IGF1R, Bcl2, PDGFRA, and caspase [56,90,91] (Figure 3). When miR-128 is downregulated, cell duplication and reduced apoptosis are favored. In addition to its role in carcinogenesis, miR-128 permits us to distinguish between high- and low-grade gliomas; in particular, it has been reported how its low expression can be associated with high-grade glioma cell lines and, consequently, with a worse prognosis [87].

miR-137

miR-137 exerts its role as a tumor suppressor by inhibiting angiogenesis and through inhibition of the EZH2 protein, a proliferation factor [56,92]. When it is downregulated, glioma development is favored because EZH2 is consequently overexpressed, and angiogenesis and proliferation are stimulated [92] (Figure 3). Sun and colleagues found out that the expression level of miR-137 was downregulated in GBM cells, and they described how the low level of this miRNA was related to poor prognosis in GBM patients [56,92,93]. miR-137 expression could be a marker for poor prognosis in GBM patients, but in the future, it might also be a new treatment approach for GBM treatment [82]. Indeed, Sun and colleagues demonstrated how the expression of miR-137 inhibited tumor growth and angiogenesis in mouse models [93].

miR-181 family

The miR-181 family is a group of tumor-suppressor miRNAs (miR-181a, miR-181b, miR-181c, and miR-181d). miR-181a targets CD133 and BMI1, markers that are related to stemness and cell proliferation [87] (Figure 3). miR-181b causes growth inhibition, apoptosis, and invasion inhibition in glioma cells by directly targeting CCN1, an inducible growth factor that promotes the adhesion of endothelial cells [94]. The most relevant for GBM development are miR-181a and miR-181b. Indeed, they could be analyzed, in the future, in diagnostic routine for a more precise classification of high-grade gliomas [77]. The miR-181 low expression can be associated with high-grade glioma cell lines and, consequently, with a worst prognosis [87]. For what concerns miR-181c and miR-181d, which regulate, in a cluster, the WNT pathway, a paper published by our group demonstrated how low expression of miR-181c or low expression of miR-181d in combination with expression of miR-648 predicts the worst prognosis [84]. Lakomy et al. observed higher levels of expression of this miRNA in the group of patients with time to progression shorter than 6 months (*p* = 0.0010) [78].

##### Tumor-Suppressor miRNAs with Effect on Therapy Efficacy

miR-370-3p

miR-370-3p suppresses cell migration and proliferation. This function is based on regulation of the WNT signaling pathway through the stabilization of β-catenin and on regulation of FOXO1, FOXM1, and TGFβ proteins [94,95] (Figure 3). In cell lines, it has been described that, when miR-370-3p is upregulated, GBM growth is inhibited, and a longer upregulation is associated with longer survival [96]. This miRNA is not strictly adequate for the classification as high- or low-grade glioma because it can be found to be downregulated in both of these situations [96].

##### Tumor-Suppressor miRNAs Involved in GBM Biogenesis That, in the Future, Will Have a Clinical Role Through Their Inhibition

miR-7

This miRNA regulates the expression of EGFR mRNA and of different mRNAs involved in the AKT/PI3K pathway, and, as a consequence, it has an impact on cell division (Figure 3). In addition, it also regulates PKM2, which is responsible for net ATP production within the glycolytic sequence. The increase in miR-7 inhibits the glucose metabolic capacity acting on the knockdown of IGF-R, which is upstream of the AKT pathway [97].

In addition to its influence on cell division, miR-7 also regulates the growth and the differentiation of GSCs [98].

The transfection of this miRNA in the U373-MG GBM cell line resulted in significant suppression of EGFR mRNA and protein, leading to the inhibition of cells’ duplication. This permits us to affirm that, in the future, after tests on tumor tissues and trials including GBM patients, the treatment with this miRNA, if found to be overexpressed, will inhibit the extension of the GBM tumor [35]. Until now, no data have been reported concerning this.

miR-34

miR-34 regulates many genes, including Bcl2; NOTCH and NUMB (involved, respectively, in apoptosis and in the development of the nervous system); CDK6; EGFR; and c-Met (which are involved in cell proliferation and invasion) [99,100,101] (Figure 3). The inhibition of these genes by miR-34 leads to cell proliferation and to a block of apoptosis, causing the development of glioma.

The expression of miR-34 can be used to treat GBM: indeed, a possibility of transmission through viral vectors and extracellular vesicles has been demonstrated [102]. In this field, good results have been obtained by Francipane and colleagues. This research group focused their attention on the use of Zika virus (ZIKV) in a GBM clinic. In GSC cultures, the infection by ZIKV induced miR-34 expression, inhibiting the anti-apoptotic protein Bcl-2 and Numb, the antagonist of Notch, both involved in GSC invasiveness. Similar data have been obtained in mouse models where ZIKV reduced brain tumor size [103,104] and metastasis [105]. The infection by ZIKV also has the advantage of being able to persist for months in blood after the initial infection, thus lowering the risk of tumor recurrence and the need for repeated viral infusions. Another fact associated to ZIKV is the enhanced effector/memory CD4+ T-cell response after the transfection, suggesting that, in the future, this virus could be a potential adjuvant to vaccine-based immunotherapies against GBM [106].

**Table 1 ijms-26-02723-t001:** List and characteristics of oncomiR in GBM.

miRNA	Target	Expression in GBM	Function/Role in GBM if the miRNA is Overexpressed	Clinical Applications		References
				In Vitro	In Vivo	
miR-9	-RAS and MYC -PTCH1	↑	Cancer cell proliferation↑ Tumor cell transformation↑ Inflammation↑ Angiogenesis↑ Apoptosis↓	Overexpression can enhance the resistance to TMZ in GBM cells (*p* < 0.05).	Associated with short-term survivors.	[51,52,53,54,55]
miR-10a, b	-CDKN1A, BIM, BCL2, TEAP2C, and PTEN -HOXD10	**↑**	Cancer cell proliferation**↑** Tumor cell migration**↑** Invasion**↑** EMT promotion**↑** Apoptosis**↓** GSC differentiation**↑**		The expression is higher in GBM than in other gliomas.	[56,57,58,59,60,61,62]
miR-17-92 cluster	-Some cell-cycle inhibitors, such as PTEN and CDKN1A	**↑**	Cancer cell proliferation**↑** Apoptosis**↓** GSC differentiation**↑**	This cluster is associated with high aggressiveness, higher invasion, and replication capability.		[48,49,50]
miR-21	-PTEN, p53 -EGFR, Cyclin D1, and AKT2 -SPOCK1 -RECK and TIMP3	**↑**	Cancer cell proliferation**↑** Tumor cell migration**↑** Invasion**↑**		Hypothetically that silencing of this miRNA can be used, in the future, as a therapy in the treatment of GBM.	[35,48,56,59,86]
miR-26a	-PTEN, ATM	**↑**	Cancer cell proliferation**↑** Invasion**↑**	Overexpression, reducing DNA repair ability and enhancing radio sensitivity to radiotherapy.		[57,80,81,82]
miR-148a	-CADM1 -FIH1	**↑**	Angiogenesis**↑** Invasion**↑**		Overexpressed in the plasma from the serum of GBM patients if compared to healthy cases. -Expressed in the high-risk group (i.e., patients characterized by low survival).	[56,64,65,66]
miR-182	-BCL2L12 -HIF2A -MET -CYLD -LRRC4	**↑**	Uncontrolled cell proliferation**↑** Apoptosis**↑** GSC differentiation**↑**		Correlates with better response to TMZ based chemotherapy and with better survival (*p* = 0.01/*p* = 0.04).	[68,69,70]
miR-196a, b	-HOXB8, HOXC8, HOXD8, HOXA7, HOXB7 -ERG -HMGA2 -ANXA1	**↑**	Cancer cell proliferation**↑** Apoptosis**↓**	Overexpression favors cells’ proliferation.	miR-196b expression correlated with OS (*p* = 0.01).	[75,76,77,78,79]
miR-648	-MOBP	**↑**	Cancer cell proliferation**↑** Invasion**↑**	The expression by transfection enhanced responsivity of TMZ in MGMT-expressing T98G glioma cells.	Correlation between OS and miR-648 expression.	[79,83,84,85]

Abbreviations: EMT, epithelial–mesenchymal transition; GBM, glioblastoma; GSC, GBM stem cell; OS, overall survival; TMZ, temozolomide; ↑, overexpressed, ↓, downregulated.

**Table 2 ijms-26-02723-t002:** List and characteristics of tumor-suppressor miRNAs in GBM.

miRNA	Target	Expression in GBM	Function/Role in GBM if the miRNA Expression is Inhibited	Clinical Applications		References
				In Vitro		
miR-1	-Connexin-43 -G6PD	↓	Cancer cell proliferation↑ Tumor cell migration↑ Apoptosis↓	Inhibition can enhance the cells proliferation and the sensitivity of GBM cells towards TMZ.		[56,87,88,89]
miR-7	-EGFR, AKT/PI3K pathway -PKM2	**↓**	Cancer cell proliferation**↑** GSC differentiation**↑**	Transfection in U373-MG GBM cell line resulted in significant suppression of EGFR mRNA and protein, leading to the inhibition of cells’ duplication.		[35,97,98]
miR-34	-Bcl2, NOTCH, NUMB -CDK6 -EGFR -c-Met	**↓**	Cancer cell proliferation**↑** Apoptosis**↓** Invasion**↑**	-In GSC cultures, the infection by ZIKV induced miR-34 expression, inhibiting the anti-apoptotic protein Bcl-2 and Numb, involved in GSC invasion. -In mouse models, ZIKV reduced brain tumor size and metastasis.		[99,100,101,102,103,104,105,106]
miR-128	-WNT -ERK -EGFR -IGF1R -Bcl2 -PDGFRA -Caspase	**↓**	Cancer cell proliferation**↑** Apoptosis**↓**	The low expression can be associated with high-grade glioma cell lines and, consequently, a worse prognosis.		[56,90,91]
miR-137	-EZH2	**↓**	Cancer cell proliferation**↑** Apoptosis**↓** Angiogenesis**↑**	Expression level of miR-137 was downregulated in GBM cells.	The low level of this miRNA was related to poor prognosis in GBM patients.	[56,92,93]
miR-181 family	-CD133 and BMI1 CCN1	**↓**	Cancer cell proliferation**↑** GSC differentiation**↑** Apoptosis**↓** Invasion**↑**		-Low level of expression was related to poor prognosis in GBM patients. -Low level of expression of miR-181c or low expression of miR-181d, in combination with expression of miR-648, predicts the worst prognosis.	[78,84,87,94]
miR-370-3p	-WNT -FOX01, FOXM1 and TGFβ.	**↓**	Cancer cell proliferation**↑** Invasion**↑**	When miR-370-3p is upregulated, GBM growth is inhibited.		[95,96]

Abbreviations: GBM, glioblastoma; GSC, GBM stem cell; TMZ, temozolomide; ↑, overexpressed, ↓, downregulated.

### 5.4. Exosomal miRNAs and Cancer

Exosomes represent a new type of material with increased relevance in the research on cancer. These vesicles derive through exocytosis, and their role is to transfer biological signals between local or distant cells. The phenomenon of exosome secretion is involved in both physiological and pathological processes. As for other biomolecules (DNA and protein), even miRNAs can be found in exosomes, at both tissue and liquid biopsy levels. Their presence has been described mainly in the lumen of these vesicles [107]. The miRNAs included in exosomes are carried from the donor cell to the recipient cell with exosomes that can transfer the contents to both nearby and distant cells. In the latter case, the exosomes move precisely through the circulatory stream. The process of sorting miRNAs in exosomes from parent cells is not a random process. The correct type and number of miRNAs are sorted in exosomes on the basis of the biological process that the parent cell needs to regulate. Guduric-Fuchs et al. and Ohshima et al. demonstrated how specific cell lines of gastric and colorectal cancers present a defined subset of miRNAs (e.g., miR-150, miR-142-3p, miR-451, and let-7 miRNA family members) in the exosomes [108,109]. In particular, Oshima et al. found that let-7 miRNA is more abundant in gastric cell lines than in lung cancer [48]. Interesting initial data have been reported concerning GBMs. In particular, it has been reported that miR-21 presented lower levels of expression in exosomes derived from serum of healthy donors compared to the exosomes derived from serum of patients affected by GBM [110]. These data create the basis, in the future, to use miRNA exosome detection in blood as a potential noninvasive biomarker to indicate disease state and stage. This concept has not yet been explored in depth in the literature.

## 6. Conclusions

The knowledge about miRNA involvement in tumor development, including GBM, has increased dramatically over the last decade. The miRNAs described in this review are the most important examples of how these non-coding RNAs can influence the development, prognosis, and efficacy of TMZ-based treatments in GBM. The information reported here shows that the incorporation of miRNA, if not all at least those for which the clinical role is quite universally accepted, into the clinical setting may provide GBM patients with a more accurate description of the treatment or prognosis of their tumor. As a consequence, the evaluation of these miRNAs could improve treatment and the definition of prognosis.

Further studies will be needed in the future to better define how these markers can be used for tumor treatment by amplification of their expression in the case of tumor-suppressor miRNAs or by their inhibition in the case of oncomiR (Table 1 and Table 2).

In addition, future research should focus more on studies involving patients than in vitro studies, because very often the data generated via in vitro experiments are not translated into so something of concrete clinical significance. Our idea is that miRNAs should not be analyzed as a single entity; instead, more emphasis should be given to their action as groups of miRNAs that collaborate in the response to therapies and prognosis of GBMs.

This is because the treatment focused on miRNAs could be a new starting point to improve the survival and quality of life of a very severe neoplastic disease, GBM, possibly by designing specific miRNAs-based therapies.

## Figures and Tables

**Figure 1 ijms-26-02723-f001:**
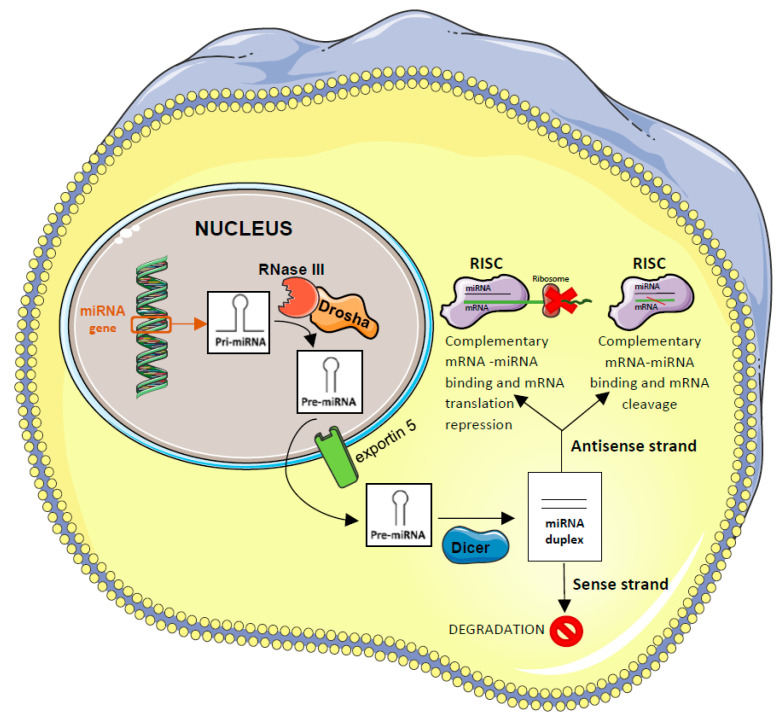
Pathway of miRNA biogenesis and action. For a specific description of the pathway, please refer to the text of the review. The figure was created by applying images reported at https://smart.servier.com/.

**Figure 2 ijms-26-02723-f002:**
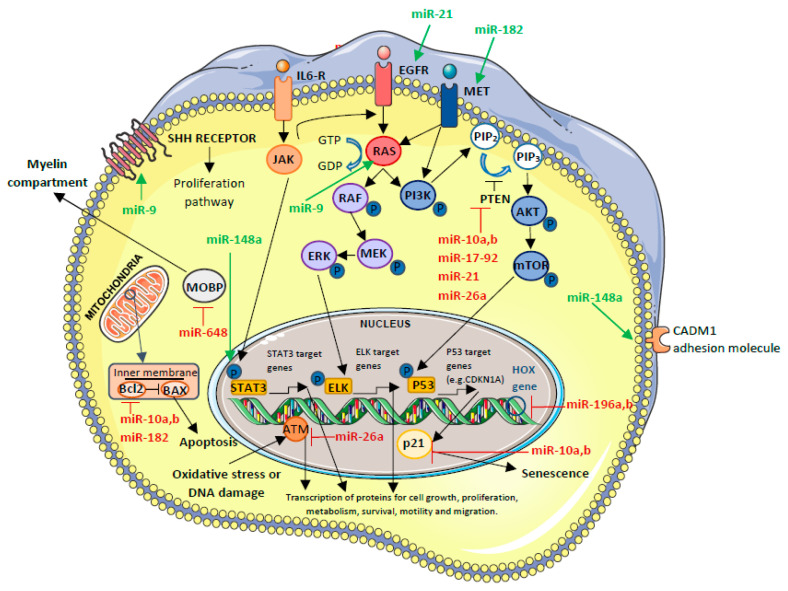
Targets of oncomiR in glioblastoma. miRNAs that block a specific pathway are shown in red, and those that activate a pathway are in green. For a specific description of the pathways reported in this figure, please refer to the text of the review. The figure was created applying images reported at https://smart.servier.com/.

**Figure 3 ijms-26-02723-f003:**
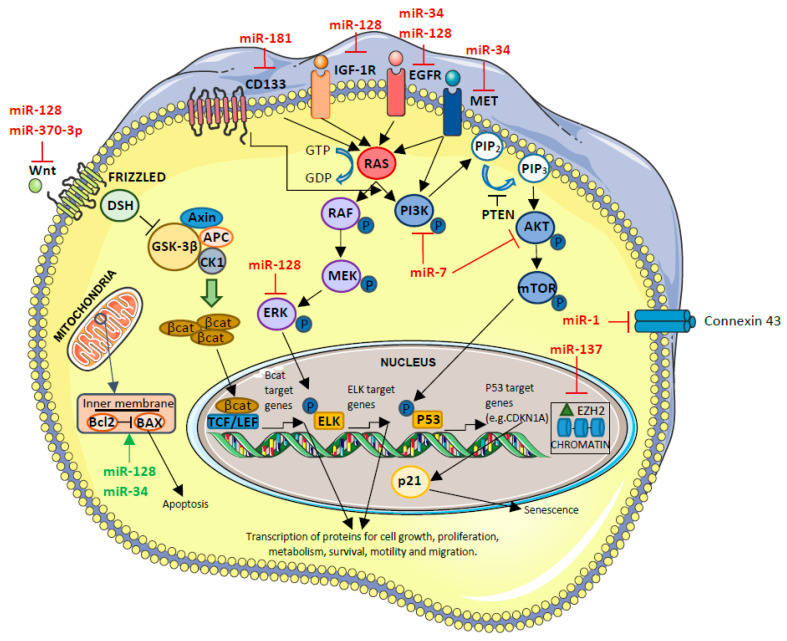
Targets of tumor-suppressor miRNAs in glioblastoma. miRNAs that block a specific pathway are shown in red, and those that activate a pathway are in green. For a specific description of the pathways reported in this figure, please refer to the text of the review. The figure was created by applying images reported at https://smart.servier.com/.

## Data Availability

No new data were created.

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
