# Peer review of "The Clinical Role of miRNAs in the Development and Treatment of Glioblastoma"

_ijms, 2025, doi:10.3390/ijms26062723_

Round 1
Reviewer 1 Report
Comments and Suggestions for Authors
The review paper “The Clinical Role of miRNAs in the Development and Treatment of Glioblastoma” by Samantha Epistolio and coworkers describes the recent applications of specific miRNAs in the development, the prognosis and the response to therapies for patients affected by GBM. This review provides an exhaustive and sufficiently updated analysis of the topic under investigation. It is well-written and clear; therefore, it is suitable for publication in this journal. I believe the manuscript can be accepted after the following comments are addressed.
A paragraph dedicated to an in-depth analysis of the applications of exosomal miRNAs in glioblastoma would make the review more complete.
Minor comments:
- increase the Figure 1 image quality;
- in figure 2 and 3 I would suggest using a more intense green for the miRNAs so indicated;
- the abbreviation TMZ should be inserted in paragraph 3, i.e. the first time temozolomide is cited;
- in the paragraph 5.1. miRNAs function and biogenesis “that exert” should be instead of “that exerts”.
Author Response
Dear reviewer,
thank you for your comments and suggestions. Please find attached in a separate file the point by point response.
Kind regards
Epistolio Samantha

Reviewer 2 Report
Comments and Suggestions for Authors
This manuscript provides a comprehensive overview of miRNAs in glioblastoma multiforme (GBM), but several issues must be addressed before publication.
Major comments:
- The literature review seems incomplete, with several key studies from the past 2-3 years missing. The authors should update their references to include more recent work.
- The methodology for selecting which miRNAs to focus on is unclear. The authors should explain their selection criteria for including specific miRNAs in their review.
- There are inconsistencies in how the clinical relevance of certain miRNAs is presented. For instance, the clinical significance of miR-182 is described as both favorable and unfavorable in different sections.
On page 11 (page 10 of the PDF), the authors state that "the expression of miR-182 correlates with better response to TMZ based chemotherapy and, as a consequence, with better survival." They cite work by Zhao et al. and Kouri et al. suggesting that miR-182 is "detected at high levels in patients with better survival" and attribute this to "the pro-apoptotic effect and to the downregulation of cancer cells migration."
However, in the same paragraph, the authors then contradict this by stating: "On the other side, some research groups have described the effect of miR-182 on growth promotion and on migration and, consequently, have associated its expression with worse prognosis." They specifically note findings that "the cumulative 5-year overall survival rate is lower when miR-182 expression is high than in the low miR-182 group for what concerns overall survival (OS) (p=0.003) and disease-free survival (DFS) (p=0.006)."
This direct contradiction within the same section presents conflicting information about miR-182's role in GBM prognosis without adequately explaining or reconciling these opposing findings. The authors should address this inconsistency by providing a more nuanced discussion of the context-dependent roles of miR-182, potential methodological differences between studies, or other factors that might explain these contradictory observations.
- The authors frequently mention findings from "in vitro" studies without addressing limitations in translating these to clinical settings.
- Tables 1 and 2 need refinement - some entries lack sufficient detail on sample sizes, statistical significance, or specific clinical parameters.
Minor comments:
- Severa errors throughout the manuscript need correction. For example,
The manuscript inconsistently capitalizes miRNA names throughout the text, which creates confusion and indicates poor editorial oversight.
For example: Other examples include inconsistent styling of "miR-7" vs "MiR-7" and "miR-128" vs "MiR-128" , miR-34” or “miR-370-3p”
- Some figure legends lack adequate detail to be interpreted independently.
- The conclusion section is overly general and does not provide specific directions for future research.
The manuscript has potential value as a review but requires substantial revision to improve scientific rigor and clarity. I recommend major revisions before it can be considered for publication.
Author Response
Dear reviewer,
thank you for your comments and corrections. Please find attached in a separate file the point by point response.
Thank you
Kind regards
Samantha Epistolio
